# Spontaneous dynamical disordering of borophenes in $MgB_2$ and related metal borides

Sichi Li [1✉], Harini Gunda [2,3], Keith G. Ray [1], Chun-Shang Wong [2], Penghao Xiao [1],
Raymond W. Friddle [2], Yi-Sheng Liu [4], ShinYoung Kang[1], Chaochao Dun [5], Joshua D. Sugar[2],
Robert D. Kolasinski[2], Liwen F. Wan[1], Alexander A. Baker[1], Jonathan R. I. Lee [1], Jeffrey J. Urban [5],
Kabeer Jasuja [3], Mark D. Allendorf [2], Vitalie Stavila [2✉] & Brandon C. Wood [1✉]

Layered boron compounds have attracted significant interest in applications from energy storage to electronic materials to device applications, owing in part to a diversity of surface properties tied to specific arrangements of boron atoms. Here we report the energy landscape for surface atomic configurations of $MgB_2$ by combining first-principles calculations, global optimization, material synthesis and characterization. We demonstrate that contrary to previous assumptions, multiple disordered reconstructions are thermodynamically preferred and kinetically accessible within exposed B surfaces in $MgB_2$ and other layered metal diborides at low boron chemical potentials. Such a dynamic environment and intrinsic disordering of the B surface atoms present new opportunities to realize a diverse set of 2D boron structures. We validated the predicted surface disorder by characterizing exfoliated boron-terminated $MgB_2$ nanosheets. We further discuss application-relevant implications, with a particular view towards understanding the impact of boron surface heterogeneity on hydrogen storage performance.

[1] Materials Science Division, Lawrence Livermore National Laboratory, Livermore, CA 94550, USA. [2] Sandia National Laboratories, Livermore, CA 94551, USA. [3] Department of Chemical Engineering, Indian Institute of Technology, Gandhinagar, Gujarat 382355, India. [4] Advanced Light Source, Lawrence Berkeley National Laboratory, Berkeley, CA 94720, USA. [5] Molecular Foundry, Lawrence Berkeley National Laboratory, Berkeley, CA 94720, USA. ✉email: li77@llnl.gov; vnstavi@sandia.gov; brandonwood@llnl.gov

Materials containing two-dimensional boron sheets, including metal diborides ($MB_2$), represent a rich and diverse class of materials with a potentially high degree of structural and electronic tunability[1,2]. As such, these layered materials and closely related borophene sheets, crystalline atomic monolayers of boron, have attracted interest in a wide variety of applications, including hydrogen storage[3], superconductivity[4], electrocatalysis[5–10], optoelectronics[11], and thermal[12,13] and corrosion resistance[14]. In several of these applications, the specific $MB_2$ surface properties play an outsized role in determining the overall behavior (for example, surface-dependent electrocatalytic function[15]). Recognizing this fact, several recent theoretical studies documented a library of possible two-dimensional boron sheets with differing surface stoichiometries and atomic arrangements generally derived from the parent P6/mmm structure of $MgB_2$, which has a graphitic arrangement of boron atoms[16–18].

Such studies highlight the importance of accurately assessing the surface structure of layered borides and borophenes, which is necessary for reliable predictions of surface-relevant properties. High-resolution ex situ imaging of boron surface structures in these materials is extremely challenging due to the low surface population of clean basal boron planes, which have relatively high surface energies[19–22] and are susceptible to contamination from surface oxidation[23,24]. Nevertheless, there is reason to believe that clean boron surfaces are manifested, particularly when boron surfaces are formed under in situ conditions. For instance, Li et al.[25] confirmed a loss of oxidized borate signal from X-ray photoemission spectroscopy of $FeB_2$ following surface hydrogen evolution, suggesting Fe surfaces are leached by the electrolyte under reaction conditions to expose the boron surface underneath. Another recent study by Sugimoto et al.[26] confirmed boron-rich surfaces of freshly cleaved $MgB_2$ sheets using low-temperature surface-sensitive scanning tunneling microscopy. Under such conditions, however, surface structure cannot be straightforwardly assessed. In particular, although it is undisputed that planar boron sheets exhibit graphitic patterns of extended hexagonal rings inside bulk crystals of metal diborides, it is an open question whether these same patterns are always preserved for boron on external surfaces. In addition to complicating the construction of reliable surface models, this knowledge gap jeopardizes understanding of how such boron surfaces behave under application-relevant conditions.

There have been some computational studies hinting at potential boron surface reconstruction in layered metal borides. Suehara et al.[16] observed from ab initio molecular dynamics (AIMD) that the stoichiometric B-terminated (0001) surface of $ZrB_2$, when annealed, exhibited boron aggregation and formation of umbrella-like $B_7$ clusters, suggesting the existence of competing stable surface patterns. Similarly, based on AIMD simulations, Kim et al.[27] predicted that the B surface of $MgB_2$ rapidly reconstructs upon adsorption of oxygen at 600 K, eliminating the $E_{2g}$ phonon mode coupling to the in-plane electronic states and disabling the high-$T_c$ superconductivity. Reconstruction has also been reported for other two-dimensional boron systems. For instance, Liu et al.[18] found by density functional theory (DFT) calculations that a free-standing boron sheet with an equivalent B density as $MB_2$ abandons the hexagonal lattice and undergoes self-amorphization via small perturbations in its initial structure[18]. This same behavior was observed for a borophene layer coated on silver substrates. Collectively, these studies suggest that the expected extended hexagonal pattern in certain exposed 2D boron sheets may not always be favored depending on the nature of the substrate.

In this work, we revisit the question of surface boron structure by combining DFT calculations with an exhaustive global optimization approach and free energy sampling. Using $MgB_2$ as a model metal diboride, we explore the complex interplay between surface electronic structure and atomic rearrangement to show that, contrary to conventional assumptions, exposed boron surfaces spontaneously and dynamically disorder. Additional electronic structure analysis reveals that the disordering tendency can be attributed to partial charge transfer at the under-coordinated surface, which leads to a frustrated, elemental-like boron state. This predicted disordering is confirmed using a combination of surface-sensitive characterization techniques applied to synthesized $MgB_2$ nanosheets. To illustrate the implications of the surface disordering, we focus on the technological application of $MgB_2$ for hydrogen storage, showing how the introduction of heterogeneity impacts hydrogenation thermodynamics and kinetics. We then apply our insights to define design rules for deliberately prohibiting or creating boron surface disorder by surveying a diverse set of layered metal diborides, thereby enhancing the understanding of boron surface chemistry and expanding the library of possible 2D boron structures.

## Results and discussion

Depending on the boron chemical potential, addition or depletion of surface boron atoms can result in a variety of different patterns. Several of these were previously explored in first-principles calculations reported by Liu et al.[18] in the context of chemical-vapor deposition synthesis, which corresponds to high relative chemical potential. Here, we are particularly interested in lower boron chemical potentials, in which the surface boron density is constrained to the stoichiometric composition. These conditions are broadly representative of exfoliation processes, as well as reaction conditions such as hydrogenation, for which there is no additional native boron source. We begin with a discussion on the possible ways in which the B atoms can rearrange within a monolayer of surface boron sheet in $MgB_2$ (shown in Fig. 1). Starting with an unreconstructed $2 \times 2$ B-terminated (0001) slab model with the expected borophene with hexagonal pattern, we first relaxed the geometry to confirm that this structure can be classified as a metastable local minimum. Next, to initiate the search for other local-minimum boron patterns, we enumerated possible surface boron atom ($B_{surface}$) locations on either bridge or hollow sites, ultimately obtaining 135 possible initial patterns for the $2 \times 2$ slab model. These structures were then relaxed to 16 unique local minima. A detailed schematic of model construction and surface structures of all local minima can be found in Supplementary Note 1, Supplementary Figs. 1 and 2. Attempts to similarly distort subsurface boron atoms had no effect, as they relaxed back to the same hexagonal pattern, thus confirming that the new local minima are found in the surface boron layer only.

Figure 1 reports the DFT-computed energies of the local-minimum surface boron structures (blue circles), referenced to the unreconstructed hexagonal borophene pattern. The values are plotted against the average B–B coordination number of $\overline{CN}_{B-B(surface)}$ defined within a cutoff distance of 2 Å and averaged over all $B_{surface}$. It is immediately apparent that patterns with lower energy than the hexagonal arrangement exist, with six such patterns identified from our initial search. We further note that $\overline{CN}_{B-B(surface)}$ for these lower-energy patterns tend to be higher (up to a maximum of 4, compared with 3 for the unreconstructed hexagonal borophene), indicating a degree of $B_{surface}$ clustering.

Top-view structures of the two iso-energetic surface boron patterns with the lowest energy from our initial search (LM2, LM3, where LM stands for local minimum) are shown alongside the pristine borophene (LM1) in Fig. 1. Changes from LM1 to LM2 correspond to migrations of two adjacent $B_{surface}$ from hollow to atop sites along their connective path, forming

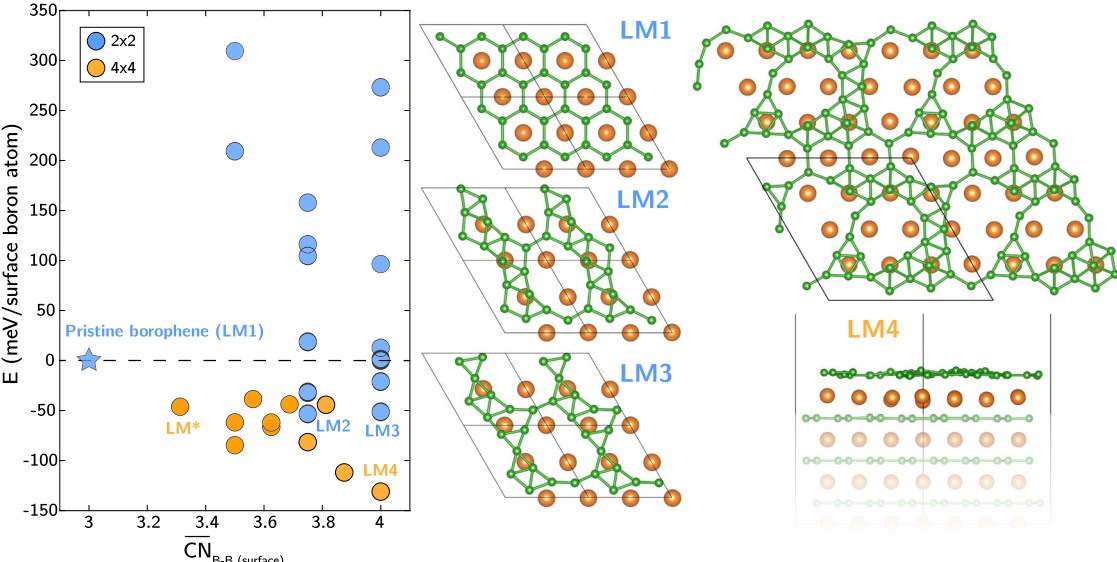

**Fig. 1 Reconstruction of surface boron sheets.** Energies of 2 × 2 (blue) and 4 × 4 (orange) slabs with different locally stable surface boron patterns found during the structure search, referenced to the pristine hexagonal borophene structure (blue star) and plotted against the average surface B−B coordination number. Select surface structures of interest are labeled, with corresponding structures shown at right (see Fig. 7 for LM* structure). The globally optimized LM4 structure is shown in top and side views at far right. Atom color codes: green: B, orange: Mg.

symmetric ten-membered B rings and exposing two subsurface Mg atoms. LM3 is structurally close to LM2, containing irregular 11-membered-B rings. Because formation of these structures results in broken B−B bonds within the hexagonal sheet, we denote it as a ring-opening process.

To explore a broader range of configurational space, we next turned to a larger 4 × 4 slab model. Here, a full enumeration of $B_{surface}$ locations becomes impractical. Instead, we created initial structures based on optimized structures extracted from the 2 × 2 slab models, then employed a Basin-Hopping global optimization scheme to access new configurations. Details of the structure-generating procedures and all optimized structures can be found in Supplementary Note 2 and Supplementary Fig. 3. Relative energies of all 4 × 4 computed slab structures are reported in Fig. 1, plotted against $\overline{CN}_{B-B(surface)}$ as orange filled circles. Due to the broader configurational space, surface boron patterns captured by the 4 × 4 surface model are generally lower in energy than the 2 × 2 ones at similar $\overline{CN}_{B-B(surface)}$. This further clarifies that the conventional hexagonal borophene arrangement is merely metastable, and that it competes with a variety of possible lower-energy arrangements. Moreover, the manifold of the lowest-energy pattern decreases monotonically with increasing $\overline{CN}_{B-B(surface)}$, highlighting the clustering preferences for $B_{surface}$, achieved by abandoning extended hexagonal ring structures to form larger, asymmetrical rings and clusters. The globally optimized surface boron structure based on the 4 × 4 model, labeled as LM4 in the rightmost panel of Fig. 1, is 131 meV/ $B_{surface}$ lower in energy than the starting hexagonal borophene. In contrast to its conventionally crystalline nature, the borophene appears to be amorphous-like and highly disordered, containing a mixture of irregular 10-, 11-, 14-membered rings with clusters of boron, such as umbrella-like $B_7$, and triangular $B_6$. This behavior is similar to that reported for the annealed borophene monolayer in $ZrB_2$ by Suehara et al.[16] where boron atoms cluster and form large ring structures. Moreover, a significant fraction of subsurface Mg atoms originally masked by $B_{surface}$ are now exposed as a result of the $B_{surface}$ ring-opening process. From the side view in Fig. 1, it is also visually obvious that some $B_{surface}$ atoms in atop positions (relative to subsurface Mg) rise from the surface.

We point out that the effect of surface composition was previously studied by Liu et al. for a range of boron chemical potentials beyond the stoichiometric composition, as represented by varying the surface boron density[18]. They reported that a boron-enriched $MgB_2$ surface can adopt an ordered boron configuration, which represents the global minimum in the zero-temperature formation energy. Because our calculations in Fig. 1 did not explicitly explore different boron densities, we performed another series of calculations to explore surface patterns at other boron chemical potentials. Details are reported in Supplementary Note 3. Our results confirm that with high boron chemical potential, boron atoms will likely form ordered patterns similar to those proposed by Liu et al. On the other hand, surface disordering remains preferred at lower boron chemical potentials, including for both stoichiometric and substoichiometric (boron-depleted) surfaces. This in turn introduces the possibility that at intermediate boron chemical potentials, surfaces will segregate into locally ordered B-enriched and disordered B-deficient regions. Note that these different ranges of boron chemical potential may correspond to different reaction conditions: for example, high chemical potentials will be observed during chemical-vapor deposition (the application of interest within the study of Liu et al.), whereas low chemical potentials will dominate during processes such as exfoliation. Similarly, under reactive hydrogenation conditions (explored further below), there is no additional native boron source, and one can expect a low relative boron chemical potential. Furthermore, atomic disordering is expected to introduce excess configurational entropy to the boron surface, which may further stabilize the disordered high-vacancy states.

Figure 2a illustrates the kinetics of the borophene reconfiguration process, computed by tracing the reaction pathways of LM1 → LM2 → LM3 transformations using the climbing-image nudged elastic band method (CI-NEB). As shown in Fig. 2a, the LM1 → LM2 transition goes through a metastable local minimum (LM#) and has an overall zero-temperature activation energy of only 0.27 eV. The activation barrier associated with the LM2 → LM3 transition is even smaller, ~0.12 eV. (The full minimum energy paths (MEPs) can be found in Supplementary Fig. 5.) We

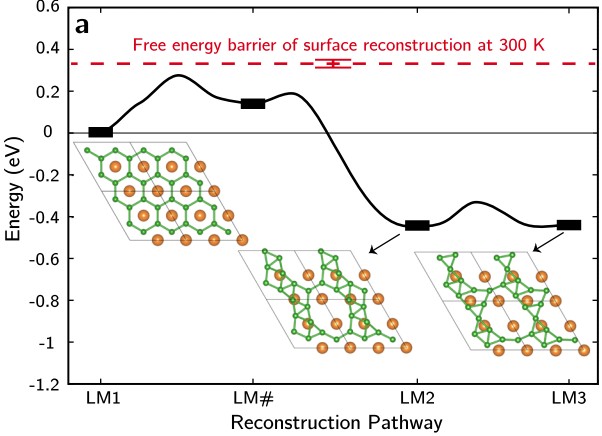

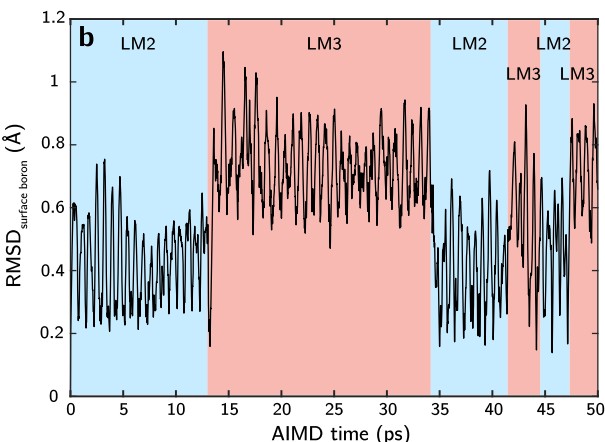

**Fig. 2 Surface boron dynamics. a** CI-NEB-computed minimum energy paths for LM1 → LM2 → LM3 conversion (black curve), along with the metadynamics-computed free energy barrier for LM1 → LM2 at 300 K (dashed red line, with error bars shown that were determined based on three replicated simulations). Atom color codes: green: B, orange: Mg. **b** Root mean squared displacement (RMSD) of $B_{surface}$ atoms from AIMD simulations of LM2 at 300 K, referenced to relaxed LM2 structure.

analysis was used to gain insights into the electronic origin of favorable boron surface reconstruction. For reference, we first determined the excess partial charges ($q$) of Mg and B in pristine bulk $MgB_2$ from Bader analysis. The average $q$ of Mg and B are +1.66 and −0.83, respectively, corresponding qualitatively to the expected formal oxidation states of +2 and −1.

Next, these bulk values were compared to the average $q$ in surface and subsurface layers of the lowest-energy 4 × 4 reconfigurations of the boron-terminated $MgB_2$ surfaces. Corresponding values for the unreconstructed $MgB_2$ surface were also included in our analysis. Figure 3a shows the average $q$ of the outermost $B_{surface}$ atoms within each computed 4 × 4 slab, plotted against $\overline{CN}_{B-B(surface)}$. The average $q$ spans a range of >0.1 $e$ across structures, with a minimum of about half of the $q$ in bulk $MgB_2$ for the hexagonal unreconstructed surface ($\overline{CN}_{B-B(surface)} = 3$). Thus, missing a layer of coordination with Mg clearly leads to partial charge depletion of $B_{surface}$. In addition, the average $q$ of $B_{surface}$ increases with $\overline{CN}_{B-B(surface)}$, reflecting further charge depletion of $B_{surface}$ upon additional surface reconstruction and boron aggregation.

Figure 3b, c break down the average $q$ of subsurface B and Mg by layer within the slab models (plotted against the average $q$ of $B_{surface}$), demonstrating the heterogeneous distribution of charge among layers. Bulk-like $q$ is attained by the second sublayer of Mg atoms and the third sublayer of B atoms (see figure caption for details). Strongly negative correlations in $q$ were found between $B_{surface}$ atoms and their closest B and Mg sublayers, suggesting the variation in the average $q$ of $B_{surface}$ across different surface configurations is mainly due to the surface-dependent charge distribution among these three layers.

Overall, our charge analyses suggest that partial charge depletion of $B_{surface}$ due to substoichiometric coordination with Mg is highly correlated with B−B bond reorganization and energetically favorable surface disordering. It is reasonable to extrapolate this concept to other oxidation scenarios and different surface terminations that may lead to similar behavior. To test this hypothesis, we created another model system to examine direct oxygen attack of a Mg-terminated (0001) $MgB_2$ surface, which mimics the oxygen susceptibility under actual operating conditions. Starting again with a 4 × 4 (0001) slab with four $MgB_2$ layers, we relaxed the structure under the addition of an oxygen monolayer on the Mg surface. Subsurface B atoms have an average $q$ of 0.38, confirming strong oxidation by surface oxygen. Global optimization via Basin-Hopping caused subsurface B atoms to undergo exothermic disordering into irregular rings and clusters very similar to the B-terminated clean surface, thus validating our hypothesis (see Supplementary Fig. 7 for surface structures of locally optimized and globally optimized configurations with oxygen). We also repeated the optimization process for a slab with the Mg surface covered by atomic H (surface hydrides with 1:1 Mg/H ratio), where subsurface B have an average $q$ of −0.81 similar to bulk B, and no subsurface B reconstruction was observed. We therefore expect that only strong oxidizing agents capable of substantially depleting the charge of the subsurface B layer would facilitate spontaneous boron layer disordering.

It is worth pointing out that charge depletion of the B layer at the $MgB_2$ surface causes these atoms to approach their elemental (neutral) charge state. This provides a clue for explaining the disordering tendency. Specifically, Ogitsu et al. reported through a global-phase space search that elemental β-rhombohedral boron contains a macroscopic amount of intrinsic defects, leading to a degenerate and disordered ground state[28,29]. The authors were able to trace this behavior to geometrical frustration of partially occupied site configurations, which mimic frustrated spin-lattice

also quantified the free energy of activation needed for the system to escape the LM1 state at a finite temperature using metadynamics simulations. These simulations reveal a free energy barrier of 0.33 (±0.02) eV at 300 K for conversion to LM2—slightly higher than the corresponding 0-K barrier—confirming that LM2 is the nearest local minimum to LM1 upon surface reconstruction (further details can be found in Supplementary Note 4). Accordingly, the kinetics of borophene disordering are readily accessible even at room temperature. The low barrier in Fig. 2a implies the likelihood that the system can interchange easily between LM2 and LM3. To confirm this, we performed AIMD simulations at 300 K. Figure 2b shows the dynamics of a reconstructed boron surface by monitoring the root mean squared displacement (RMSD) of $B_{surface}$ atoms. Multiple episodes of LM2 ↔ LM3 interconversion are detectable even within these short simulation timescales at room temperature, manifesting the highly dynamic nature of the boron surface. Collectively, the results point to a $MgB_2$ surface structure that disorders spontaneously and dynamically under conditions of low boron chemical potential, in sharp contrast to the assumption of a static hexagonal arrangement.

The surprising dynamical disordering tendency of the boron surface in $MgB_2$ can be understood by analyzing the surface electronic properties in detail, as shown in Fig. 3. Bader charge

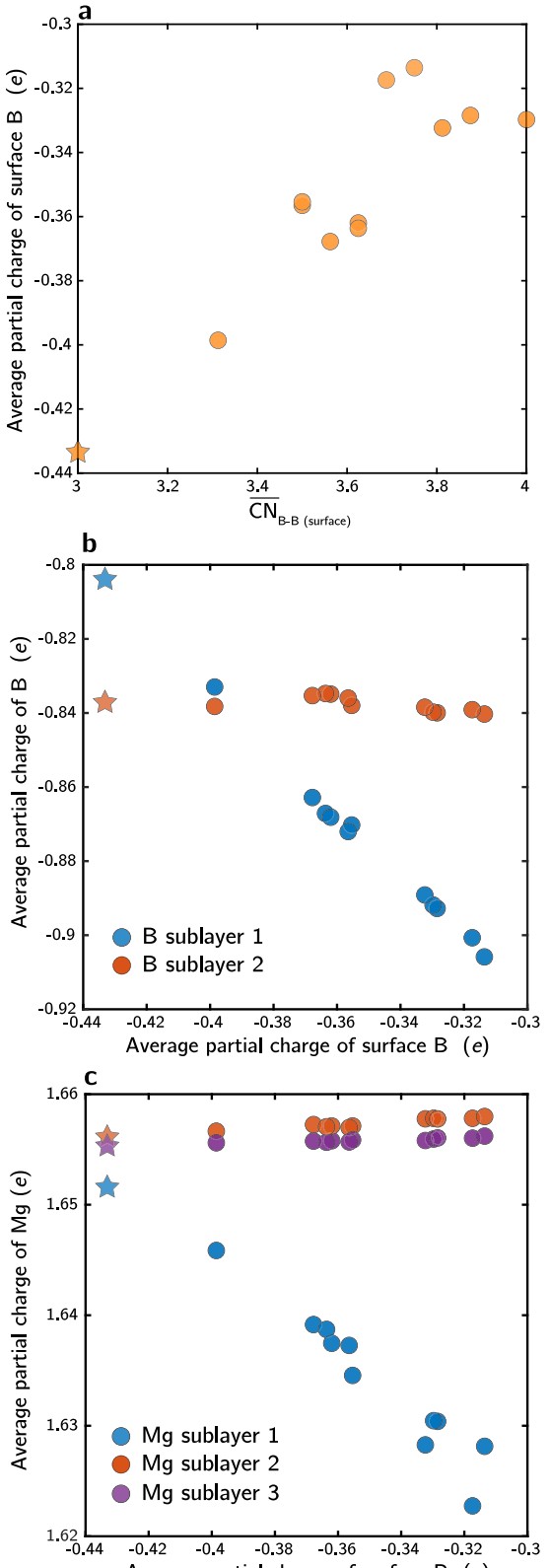

**Fig. 3 Partial charge analysis. a** Average partial charge of surface B atoms as a function of surface B−B coordination number. Average partial charge of **b** other B layers and **c** Mg layers as a function of the average partial charge of surface B atoms. B and Mg layer indices are defined according to their proximity relative to the outermost boron surface. Data are compiled from all computed 4 × 4 surface configurations in Fig. 1. Stars represent the pristine hexagonal surface.

configurations and are intrinsic to elemental boron. Similarly, Liu et al. found that a free-standing charge-neutral boron sheet in vacuum strongly prefers disordering and forms amorphous-like patterns. In the case of $MgB_2$, we propose that formation of elemental-like surface B due to substoichiometric coordination with Mg drives $B_{surface}$ into an electronic state close to elementary frustrated B substance, which introduces a degree of electronic frustration that manifests as spontaneous dynamical disordering.

To validate the predicted disordering of $B_{surface}$ atoms, we synthesized $MgB_2$ nanosheets (NSs) with predominant boron surface termination through mechanical exfoliation using high-energy ball milling, followed by dispersion in acetonitrile and subsequent centrifugation (see "Methods")[30]. Using transmission electron microscopy (TEM) and atomic force microscopy (AFM), we determined that the exfoliated $MgB_2$ NSs were flat with lateral dimensions ranging from 100 to 500 nm and thicknesses of ~4 nm (Fig. 4a and Supplementary Fig. 8). The dominant $MgB_2$ composition was confirmed using energy-dispersive X-ray spectroscopy (EDS) analysis (Supplementary Fig. 8). To comment on the crystalline nature and the degree of exfoliation of $MgB_2$, we acquired X-ray diffraction (XRD) data of bulk and exfoliated $MgB_2$ NSs (Fig. 4b). The exfoliated $MgB_2$ NSs show significant peak broadening, likely due to finite size/Scherrer and/or strain effects resulting from a high degree of exfoliation of $MgB_2$. Moreover, the intensity of the (001) and (002) peaks corresponding to the C-stacking planes of $MgB_2$ is highly diminished, indicating the presence of only a few layers within the crystals [31].

In order to confirm that the $MgB_2$ sheets are predominantly B-terminated, we performed low energy ion scattering (LEIS), which can provide an elemental analysis of the outermost surface monolayer[32]. LEIS measurements of $MgB_2$ NSs pressed into an indium foil suggest a primarily boron-terminated surface, based on two main observations. First, the presence of boron at the surface was confirmed with an ion energy spectrum taken with a 3 keV $Ne^+$ beam in Fig. 4c. This spectrum contains a large B(R) peak, corresponding to detection of surface boron being recoiled by incident $Ne^+$ into the detector. Second, to interrogate the boron surface coverage, LEIS measurements were taken with a 3 keV $He^+$ beam. These spectra, plotted in Fig. 4d, reveal a strong matrixing effect (neutralization of the incident ions). All higher-energy peaks associated with scattering from all surface species (B, O, Mg, and In) are greatly suppressed. For comparison, the blue dotted line shows a spectrum for a clean Mg surface, which contains a large Mg(QS) peak. Similar matrixing effects have been reported for complete graphitic carbon monolayers on metal surfaces by Mikhailov et al.[33], who speculated that these effects might also occur for other light elements strongly bonded to metals. The combination of these two observations demonstrates that boron is present at the surface and likely constitutes a large portion of the surface termination.

The native disordering of the boron planes near the surface was probed using X-ray absorption spectroscopy (XAS) measurements, which was previously established as a reliable technique for assessing the intactness of the boron ring structure[24]. In particular, a peak near ~187 eV corresponds to B $2p_{xy}$ states that are characteristic of the B−B hexagonal ring structure of bulk $MgB_2$[34]. XAS measurements of the exfoliated $MgB_2$ NSs were performed in both TEY (total electron yield, a surface-sensitive mode) and TFY (total fluorescence yield, a more bulk-sensitive mode) modes. As shown in Fig. 5a, the B−B ring feature is indeed completely missing in the TEY mode in the $MgB_2$ NSs. The feature at 193.7 eV corresponds to $B_2O_3$ (or other substoichiometric boron oxides), which indicates that some limited oxidation is occurring even in these pristine samples. However, the exfoliated $MgB_2$ NSs also show small peak-like features between 191.5 and 193 eV, which are not present in the $B_2O_3$, and similar

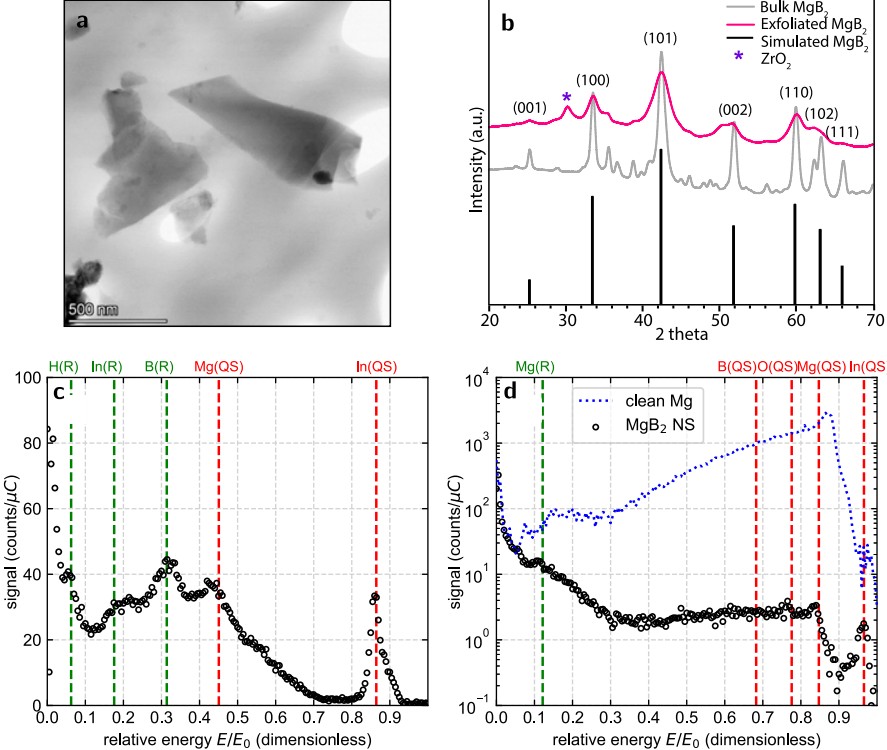

**Fig. 4 Physical and chemical characterization of the exfoliated MgB₂ NSs. a** Representative TEM image of the exfoliated MgB₂ NSs. **b** XRD analysis of bulk or standard MgB₂ and exfoliated MgB₂ NSs, compared with simulated MgB₂ diffraction data. A small peak at 30.4° corresponds to zirconia (ZrO₂) contamination obtained during high-energy ball milling (HEBM) of MgB₂. **c** Ion energy spectrum for 3 keV Ne+ on the MgB₂ NSs for $\alpha = 76°$ and $\theta = 54°$. Scattering peaks are labeled X(QS) in red, while recoil peaks are labeled X(R) in green, to denote scattering or recoiling a surface atom of species X. The ion energy spectrum reveals a sizeable B(R) peak, evidence that boron is at the MgB₂ NS surface. **d** Ion energy spectrum for 3 keV He⁺ on MgB₂ NSs is plotted as open black circles for $\alpha = 76°$ and $\theta = 60°$. For illustrative comparison, a second ion energy spectrum for 2 keV He⁺ on a clean Mg surface (for $\theta = 55°$) is shown as a blue dotted line. The higher-energy scattering peaks for the MgB₂ NSs are greatly suppressed by a matrixing effect. Note that a small shift in the relative energy of the Mg(QS) peaks between the two spectra is expected due to the slightly different scattering angle $\theta$.

features are observed in this range for a powder sample of elemental boron. This is consistent with our theoretical calculations suggesting that B-rich surfaces should feature complex structures with higher B coordination that are closer to neutral oxidation state, similar to those in elemental boron.

As further confirmation of our interpretation, we also directly simulated the B K-edge XAS spectra for pristine and reconstructed MgB₂ surface layers (based on the LM4 reconstruction). As the surface boron monolayers on MgB₂ are predicted to be amorphous with dynamically changing configurations, we propose that XAS is actually probing a geometric ensemble and time average of surface structures. Nonetheless, contrasting the pristine borophene and LM4 surface boron patterns should qualitatively inform the effect of atomic disordering on the spectral line shapes. The results are shown in Fig. 5b. Whereas the 2p$_{xy}$ feature near 187 eV is retained for the pristine surface, indicating intact surface boron rings, the reconstructed surface is indeed missing this feature. The reconstructed surface also exhibits increased spectral weight between 188 and 193 eV, including a small peak near 190 eV and shoulder near 191.5 eV that are reminiscent of similar features between 191.5 and 192.5 eV in the exfoliated MgB₂ B K-edge TEY XAS measurements (Fig. 5a). Thus, the XAS signature of the theoretically predicted surface reconstruction is consistent with spectrum of the exfoliated MgB₂ B K-edge TEY XAS measurements, strongly supporting the hypothesis that the disruption of the hexagonal ring structure produces substantial surface disorder. We acknowledge the possibility that ball milling

could enhance the kinetics of surface disordering within our samples. However, Sugimoto et al.[26] similarly observed through scanning tunneling microscopy that boron atoms on a freshly cleaved MgB₂ sheets are randomly distributed and irregularly arranged even in the absence of ball milling. These results confirm that surface disordering also occurs under nonaggressive synthetic conditions.

The native surface disorder introduces heterogeneity in local surface properties of MgB₂, with potentially broad implications for application performance. As one illustration, Fig. 6 explores the interaction of disordered MgB₂ with hydrogen in view of its utility as a hydrogen storage material[3]. MgB₂ can absorb hydrogen and form Mg(BH₄)₂, which has among the highest gravimetric hydrogen capacities of any metal hydride at around 14.9 wt.%. However, the MgB₂-Mg(BH₄)₂ hydrogen storage system suffers from sluggish kinetics of hydrogenating MgB₂ to Mg(BH₄)₂, necessitating high temperatures (350−400 °C) in hydrogenation half cycles. Liu et al.[34] proposed that the intrinsic stability of the B−B extended hexagonal ring structure hinders the overall hydrogenation kinetics of MgB₂ (with an underlying assumption that B maintains hexagonal patterns even when surface-exposed). As such, it is critical to understand which factors can facilitate B−B bond disruption under hydrogen-rich reaction conditions with low boron chemical potential. The current study suggests that such disruption is achievable natively at surfaces due to surface disordering, prompting a more direct investigation into the implications for hydrogenation.

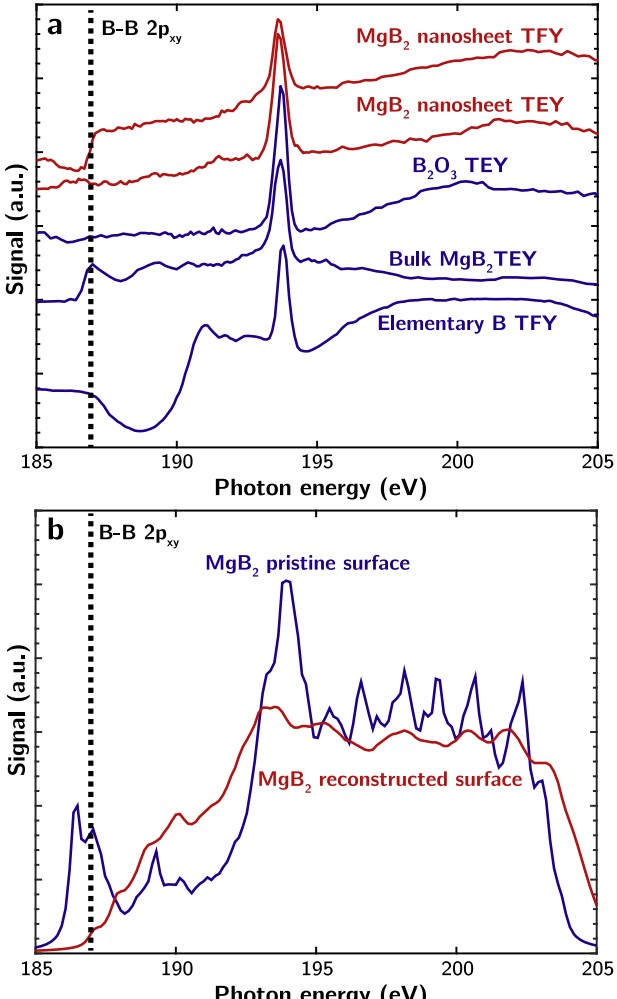

**Fig. 5 XAS measurements and simulations of exfoliated MgB₂ NSs. a** Boron K-edge TEY (total electron yield; surface-sensitive) and TFY (total fluorescence yield; more bulk-sensitive) modes of XAS spectra for exfoliated MgB₂ nanosheets, compared to reference spectra for other bulk boron-containing compounds. **b** Simulated XAS spectra for MgB₂ slabs with pristine or LM4-reconstructed boron surfaces. The decrease in the simulated signal near 203 eV is due to the finite number of electronic bands included in the calculations.

For reference, we first computed single atomic H binding to a bridge site on the unreconstructed hexagonal MgB₂ surface, finding an H binding energy of $E_{H\,binding} = -0.68\,eV$ (referenced to $H_2$ gas). To perform a similar analysis on the disordered MgB₂ surface, we enumerated all possible bridge boron sites and accessible subsurface hollow Mg sites on the LM4 configuration. Figure 6a reports all local minima identified after geometry optimizations, colored by the computed $E_{H\,binding}$. A strong site-dependent heterogeneity is observed, spanning a range from −0.5 to 0.5 eV, as highlighted in the histogram in Fig. 6b. Overall, interactions between the boron surface and H are weakened after surface disordering. Perhaps more importantly, the large range of energies indicates an especially large variety of intrinsic local reactivities.

The $H_2$ dissociation kinetics exhibit a similarly wide range of local intrinsic variability on disordered MgB₂. From zero-temperature CI-NEB, the reference surface $H_2$ dissociation barrier onto two neighboring bridge sites of the unreconstructed hexagonal MgB₂ surface is found to be 0.52 eV (full MEP can be found in Supplementary Fig. 9). Figure 6a reports the calculated

$H_2$ dissociation kinetics on three representative pairs of neighboring H binding sites on the disordered LM4 MgB₂ surface with different relative pairings of $E_{H\,binding}$ values. We also included in our analysis a site pair consisting of a subsurface Mg site and a site on surface B (detailed MEPs can be found in Supplementary Fig. 9). Dissociation barriers at the three paired-B sites are all higher than on the pristine hexagonal surface, suggestive of lower reactivity of the $B_{surface}$ alone with $H_2$ upon surface disordering. However, dissociation on the surface B and subsurface Mg sites is more facile (0.38 eV), suggesting that nanostructuring to maximize boron-surface-area-to-volume ratio is beneficial for hydrogenation kinetics of MgB₂. The transition state structure, reported in Fig. 6a, shows that as $H_2$ begins to dissociate, it stays in close contact with both $B_{surface}$ and subsurface Mg. The Lewis acid−base pair interaction created by the Mg and B (see Fig. 3a, c) polarizes the $H_2$ molecule, stabilizing the transition state structure and facilitating H−H bond cleavage. Indeed, our calculations confirm that at the transition state, the two H atoms have opposite partial charges of −0.5 and +0.05 for the Mg-bonded and B-bonded H, respectively.

As a result, exposure of subsurface Mg upon boron surface disordering introduces a new mechanism for facile $H_2$ dissociation on select sites. Under hydrogenation conditions, it is reasonable to assume that such sites could function as the predominant "scissors" for $H_2$ cleavage. As exothermic atomic H diffusion is expected to be facile, exemplified by low H diffusion barriers[35] in bulk MgB₂ and on the pristine Mg-terminated (0001) surfaces[36], the resulting atomic H can then diffuse to other nearby favorable binding sites. However, one caveat is the lower density of sites with negative $E_{H\,binding}$, which may prevent the disordered surface from attaining a high H coverage (assuming all $B_{surface}$ remain intact during hydrogenation). Accordingly, the disordered surface is likely to exhibit a competition between site activity and coverage (site density).

Inspired by our findings for MgB₂, we performed a general assessment of surface disordering tendency at stoichiometric surface composition for a diverse set of metal diborides with the P6/mmm space group that have been researched and reported in the literature, with metals ranging from s-metals (Mg, Al), first-row transition d-metals (Sc, Ti, V, Mn), late transition d-metals (Nb, Ta, Mo, Zr, Hf) and f-elements (U). Notably, Yousaf et al.[37] confirmed the feasibility of synthesizing nanosheets for most of these metal diborides through liquid phase dispersion, allowing for future experimental validation of our predictions. For each metal diboride, we simulated the modest surface distortion found for MgB₂ shown in Fig. 7a that is labeled as LM* in Fig. 1. Relaxed structures remained similar to their initial configurations for most diborides, except for FeB₂ and YB₂, which underwent significant spontaneous structural reorganization following initial distortion, as illustrated in Supplementary Fig. 10. Figure 7b reports the reaction energies ($\Delta E$) associated with $B_{surface}$ disordering for each metal diboride. The values of $\Delta E$ vary significantly across different diborides, spanning both positive and negative regimes.

Overall, we predict that ScB₂, MnB₂, MgB₂, YB₂, and FeB₂ should all exhibit native surface disordering, with FeB₂ showing the strongest reconstruction tendency. In these substances, the average partial charges of $B_{surface}$ atoms, particularly in MnB₂ and FeB₂ (−0.22 and −0.15, respectively), are all less negative than that in MgB₂ (−0.42). Beyond the energetic benefits, surface disordering also enhances configurational entropy, which may lead to additional stabilization for diborides that are slightly positive in $\Delta E$. This provides a rationalization for observations of reconstructed ZrB₂ by Suehara et al.[16]. Moreover, the linkage between metal oxidation tendency and surface disordering tendency suggests that disordering could be induced in other

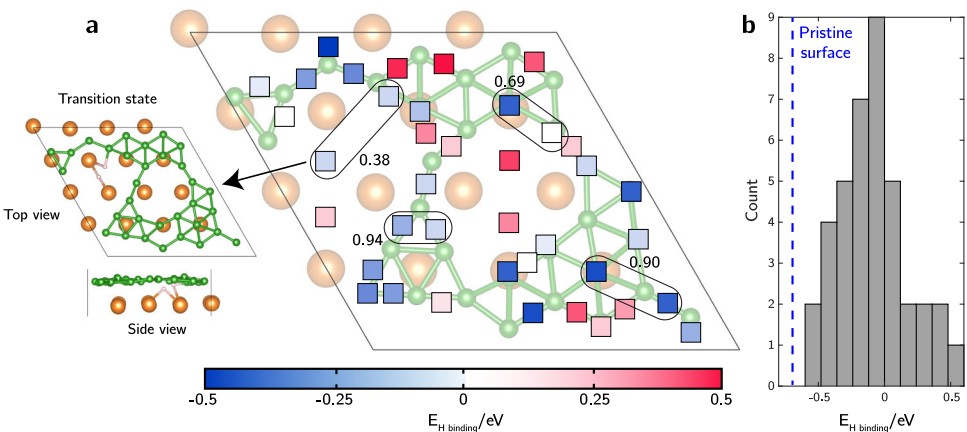

**Fig. 6 Implications for MgB₂ hydrogenation. a** Energy map of atomic H binding on the LM4 configuration of MgB₂, with squares indicating locally stable binding sites colored according to $E_{H\ binding}$. Tested site pairs for dissociative $H_2$ adsorption are highlighted as black ovals, with corresponding dissociation barriers indicated in eV. Top and side views of the transition state structure for dissociative $H_2$ adsorption at the Mg- B site pair are shown at left. Atom color codes: green: B, orange: Mg, pink: H. **b** Histogram of $E_{H\ binding}$ on the LM4 configuration compared with the reference value for the hexagonal boron surface (vertical dashed line).

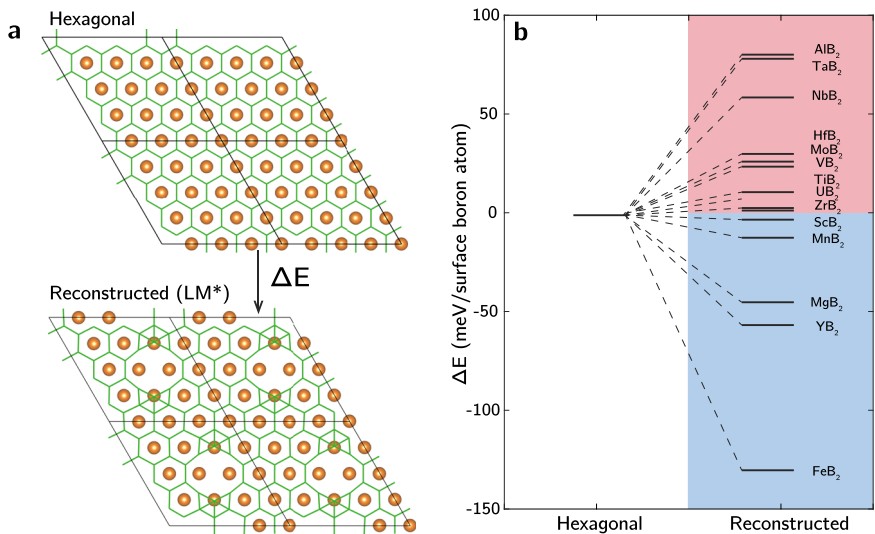

**Fig. 7 Predictions of surface disordering for other metal diborides. a** Structures of the pristine hexagonal and modestly distorted $MB_2$ boron surfaces used as initial configurations to assess surface disordering tendency. Atom color codes: green: B, orange: M. **b** Energy diagram of final boron surface configurations following structural relaxation from LM*, referenced to the hexagonal boron surface. Colors distinguish metal diborides exhibiting endothermic reconstruction (pink) from those exhibiting exothermic reconstruction (blue).

diborides by substitutional doping with elements such as Mg, Y, or Fe, or suppressed by doping with Al, Ta, or Nb, etc. It is important to note that the predictions in Fig. 7 assume a stoichiometric surface; other boron patterns may be favored under different reaction conditions[38,39]. Nevertheless, we assert that disordering should be observable within certain ranges of boron chemical potential. In addition, surface atomic rearrangements and amorphization were also observed by Guo et al.[40] for boron carbide nanosheets through Raman spectroscopy, suggesting that similar physiochemical phenomena may be relevant for a wider range of nanoscale materials.

In conclusion, our results unravel a previously unrecognized spontaneous surface disordering phenomenon of boron surfaces of MgB₂ and certain related metal diborides. This disordering, which occurs under conditions of lower boron chemical potential, can be linked to the formation of frustrated surfaces that mimic the known complex behavior of elemental boron. The behavior

was confirmed in synthesized MgB₂ nanosheets, which showed spectral evidence that is consistent with native disordering of the boron hexagonal ring structure. Moreover, this surface atomic reorganization is predicted to be dynamic, suggesting local surface bonding character in fact fluctuates.

One immediate consequence of our findings is that idealized surface models may give a misleading picture of the surface-relevant properties of metal diborides. Computational models generally assume hexagonal B surfaces for such compounds[22,25,41]; while such an assumption may hold true for some metal diborides, mechanistic interpretations for others can be inaccurate. Moreover, when accurate surface energies are required, as in the prediction of properties of nanosized metal diborides, hexagonal surface models that do not account for stabilization upon disordering may not be sufficiently representative. Likewise, when screening surface activity based on relevant computed descriptors, for instance using H binding energy to an active site to infer its hydrogen evolution

reaction activity[25], an accurate determination of surface arrangement is prerequisite.

In addition, our work points to important practical consequences. Because disordered surfaces exhibit a broader heterogeneity of local properties, this new understanding of the nature of the boron surface also has nonobvious implications for performance. One example lies in the increased richness of hydrogenation chemistry for hydrogen storage applications, for which surface disordering of $MgB_2$ was found to expose a new $H_2$ dissociation pathway that leverages Lewis acid−base pair interactions to significantly lower barriers. At the same time, it provides insight into how the stability of boron ring structures may be altered, which has been suggested as a first step towards rupturing the $MgB_2$ lattice and promoting phase transformations during later-stage hydrogenation. This highlights the necessity of considering the surface atomic disordering in interrogation of mechanisms and interpretation of experimental observations, while simultaneously guiding approaches for improvement.

Finally, we point out that the rich structural diversity of patterned borophenes has already generated intense interest due to the variety of potentially tunable properties. We suggest that natively disordered boron surfaces can extend this catalog to include amorphous variants. Indeed, our study points to a possible engineering strategy for generating surface disorder in diborides via substitutional doping of metals with higher primary oxidation states, which opens up a new avenue for exploration of amorphous 2D boron structures with a wider variety of compositions. This is particularly notable in light of recent interest in high-entropy alloys of metal diborides[42], which offer a diverse palette for altering composition. The potential tunability prompts the possibility of engineering boron surfaces of nanoscale and 2D systems with specifically tailored properties—for instance, for use in coatings, composites, electronic materials, or device applications[43,44]. Likewise, the diversity of binding configurations due to the presence of surface disorder could be an important tool for realizing multifunctional catalytic applications[31,45].

## Methods

**DFT calculations**. Plane-wave, periodic supercell DFT calculations were performed with the Vienna ab initio simulation package (VASP), version 5.4.4 [46], using the projector augmented wave treatment of core−valence interactions[47,48] with the Perdew−Burke−Ernzerhof (PBE)[49] generalized gradient approximation for the exchange-correlation energy. Initial bulk structures of metal diborides were obtained from the Materials Project[50]. Enumeration of symmetry-distinct initial configurations was performed using the enumlib code[51] wrapped in pymatgen[52]. Supercell generation was facilitated by the Atomic Simulation Environment (ASE) package[53]. To model the (0001) boron-terminated surface, we first cleaved bulk $MgB_2$ into a (0001) $2 \times 2$ stoichiometric slab with four $MgB_2$ layers, each with eight boron and four Mg atoms. Each slab was separated from periodic images by at least 15 Å of vacuum. Such slab separation by vacuum was confirmed to be sufficient by Effective Screening Medium[54] calculations implemented in Quantum Espresso[55] that showed negligible dipole−dipole interactions of slabs across successive periodic images along the z-direction. The bottom two layers containing Mg surface were fixed, and the top two layers including the boron surface are allowed to fully relax. To further sample the configurational space of boron surface atomic arrangements, we quadrupled the surface size to $4 \times 4$ to reduce translational symmetry imposed by periodic boundary conditions, maintained the slab thickness, and performed a selection of geometry optimizations and Basin-Hopping global optimizations. For all calculations, the energy cutoff for the plane-wave basis set was set to 500 eV, self-consistent-field electronic energies were converged to $10^{-4}$ eV and atomic forces to <0.03 eV/Å. Spin polarization was turned off unless specified, as most materials simulated are nonmagnetic. The Brillouin zone was sampled with a Γ-centered $4 \times 4 \times 1$ k-point mesh for $2 \times 2$ slab structures, and a Γ-centered $3 \times 3 \times 1$ k-point mesh for $4 \times 4$ slab structures. We deem such k-point sufficient as test calculations for $MgB_2$ with higher k-point density cause only negligible differences in relative energies across structures with the same lattice. The mesh of k-points was reduced to $2 \times 2 \times 1$ for climbing-image nudged elastic band (CI-NEB)[56,57] calculations for $4 \times 4$ slabs, but was maintained as $4 \times 4 \times 1$ for $2 \times 2$ slabs, allowing for direct energetic comparisons. Bader charge analyses were performed using the algorithm developed by Henkelman et al.[58–60]. For atomic H binding calculations, binding strength was quantified by referencing

to gas phase $H_2$ following

$$E_{\text{H binding}} = E_{\text{surface+H}} - E_{\text{surface}} - \frac{1}{2}E_{H_2} \qquad (1)$$

where $E_{\text{surface + H}}$, $E_{\text{surface}}$ and $E_{H_2}$ are DFT-computed energies of a surface with an adsorbed atomic H, the adsorbate-free clean surface and $H_2$ in gas phase, respectively.

**Ab initio molecular dynamics and metadynamics simulations**. NVT ab initio molecular dynamics (AIMD) were performed for $2 \times 2$ slabs with the energy cutoff for the plane-wave basis set reduced to 400 eV and the Brillouin zone sampled with a Γ-centered $3 \times 3 \times 1$ k-point mesh. The timestep was set to 1 fs, and a Nose−Hoover thermostat was used to maintain the temperature. To quantify the dynamics of boron surfaces, the root mean square displacement (RMSD) of $B_{\text{surface}}$ atoms in the course of AIMD trajectories were collected after 3 ps of pre-equilibration for each system. To compute the free energy barrier of surface reconstruction at finite temperatures, metadynamics were performed starting with the $2 \times 2$ pristine slab. Metadynamics is a nonequilibrium molecular dynamics method capable of efficiently sampling free energy surfaces of complex reactions[61]. As surface reconstruction requires B−B bond breaking of the pristine surface, we selected the coordination number of an arbitrary pair of bonded $B_{\text{surface}}$ atoms as the collective variable (CV), mathematically defined as

$$\frac{1 - \left(\frac{d_{ij}}{d_0}\right)^9}{1 - \left(\frac{d_{ij}}{d_0}\right)^{14}} \qquad (2)$$

where $d_{ij}$ is the actual distance between B $i$ and $j$, and $d_o$ is the reference distance as the boundary of being bonded or not between the two atoms, which was set to 2 Å. A total of 3 ps of AIMD were run to pre-equilibrate the system before bias potentials were introduced. Gaussian potentials with a height of 0.002 eV and a width of 0.02 were added to the defined CV every 10 fs. Following a previously reported protocol[62], we terminated a run of metadynamics simulation after the first barrier crossing from the reactant basin into the target product basin, and computed the free energy barrier by summing up the amount of bias potentials accumulated in the reactant basin. Error bars were computed on the basis of three replicas starting with different initial configurations.

**Basin-Hopping global optimization**. Global optimizations based on the Basin-Hopping algorithm[63,64], as implemented in ASE, were performed for several $4 \times 4$ slab structures for which exhaustive configurational sampling was computationally demanding. In each Basin-Hopping step, only selective atoms expected to reconstruct a posteriori were randomly displaced around the initial position following a Gaussian distribution with a variance of 0.15 Å. From there a local structure optimization was performed and the final energy was used to accept or reject this move in the Metropolis Monte Carlo algorithm. The temperature was set to 600 K which allowed systems to temporarily visit high-energy local minima in order to reach other local minima potentially barred by huge "barriers". As Basin-Hopping requires extensive local optimizations, atomic forces were only converged to 0.20 eV/Å for each step with energy cutoff for the plane-wave basis set reduced to 400 eV, and Brillouin zone sampled at Γ point only. The lowest-energy structures obtained were then re-optimized following higher-accuracy DFT parameters, energy, and force criteria described in the previous section.

**X-ray absorption spectra (XAS) simulations**. XAS simulations of the B K-edge were performed using first-principles DFT within the the Vienna ab initio simulation package (VASP), version 5.4.4 [46]. A plane-wave cutoff of 600 eV was used and the k-point sampling was chosen such that the density of k-points was ≥64,000 per $Å^{-3}$. The XAS/XES simulations were carried out in VASP using an implementation that computes the required matrix elements within the PAW formalism including a self-consistent treatment of the core-hole effects on the unoccupied states[65]. Spectra were broadened with a 0.25 eV Gaussian.

**Synthesis and preparation of $MgB_2$ nanosheets**. $MgB_2$ powder (Sigma-Aldrich, ≥99% purity, 100 mesh size) was subjected to solid-state mechanical exfoliation using a high-energy ball mill (HEBM, SPEX SamplePrep, 8000M Mixer/Mill). The process was carried out in a zirconia ($ZrO_2$) vial using 10-mm diameter $ZrO_2$ balls. Briefly, 0.80 g of $MgB_2$ powder and 20.16 g of $ZrO_2$ balls were weighed in the argon glove box (≤0.1 ppm $O_2$ and $H_2O$) and added to the milling vial to obtain a ball to powder weight ratio of 25:1. The milling vial was closed tightly in the argon glove box and then subjected to HEBM for 4 h. The milled powder was collected in a glove box and stored in a vial. To separate the exfoliated nanosheets (NSs) from the milled powder, we added 90 mL of dry acetonitrile (ACN) to 0.64 g of milled $MgB_2$ powder. We chose ACN as it has a freezing point of −45 °C and a boiling point of 82 °C, which makes it feasible to operate in the lyophilizer in the subsequent synthesis procedure. The suspension ($MgB_2$/ACN) was mixed using a bath ultrasonicator (Cole-Parmer 8891 Ultrasonic cleaner) for 5 min. The obtained dark black uniform mixture was exposed to centrifugation (Eppendorf centrifuge-5430R) at $207.53 \times g$ for 45 min to separate the heavier fraction. The top supernatant was collected and left undisturbed overnight to allow other heavier fractions to settle down. After settling, the top black

and clear supernatant was collected, which remains a stable uniform dispersion and exhibits a strong Tyndall effect. The obtained uniform dispersion of $MgB_2$ NSs in ACN was subjected to freeze-drying (lyophilization) to recover the nanosheets in the powder form. The dry $MgB_2$ powder was collected in a vial and stored in a glove box until further analysis. Freeze-drying (lyophilization) was performed by taking the dispersion of $MgB_2$ NSs in ACN (70 mL) in two 50-mL centrifuge tubes and freezing them at −80 °C overnight. The frozen samples were subsequently placed in a lyophilizer (SP Scientific, VirTis BenchTop Freeze Dryer) for up to 72 h to obtain a powder of $MgB_2$ NSs.

**Powder X-ray diffraction**. Powder X-ray diffraction measurements were recorded on an Oxford Diffraction Supernova in capillary mode using Cu Kα radiation, with a CCD detector at 77 mm from the samples and an exposure time of 66 s. The recorded 2D diffraction images were integrated to produce a 1D diffraction spectrum. The powder samples were packed in 0.7- or 0.5-mm diameter glass capillaries sealed with silicone grease under argon.

**Transmission electron microscopy**. TEM and energy-dispersive X-ray spectroscopy (EDS) were recorded on a Thermo Fisher Titan Themis Z TEM operated at 300 kV. Samples for TEM analysis were prepared by taking a small amount of $MgB_2$ nanosheet powder and dispersing it in 10 mL of dry toluene, to prevent excessive agglomeration of the nanosheets, using a bath ultrasonicator for 5 min. Two drops (20 μL) of the dispersion were drop casted on a 300 mesh copper TEM grid coated with a lacey carbon film. The grid was air-dried for a few minutes and stored in a vial until analysis.

**Atomic force microscope (AFM)**. AFM images were acquired with a Dimension ICON (Bruker) contained in an argon glove box. The AFM was operated in Peak-Force Tapping mode with a nominal tapping force of 1 nN. The tip used was a PFQNE-AL (Bruker) with spring constant of 0.84 nN/nm. Samples for AFM analysis were prepared by static dispensing of the $MgB_2$ NSs dispersion (~20 μL) on a cleaned and UV Ozone treated Si/SiO2 substrate and spin-coated at a speed of 3000 rpm for 30 s at 100 ramp rate. The coated substrate was kept in a box and stored under an argon environment until analysis.

**Low energy ion scattering (LEIS) and direct recoil spectroscopy (DRS)**. The surface termination of the $MgB_2$ nanosheets was investigated with LEIS and DRS. These measurements were performed with an angle-resolved ion energy spectrometer (ARIES), which has been described in detail in previous works[66,67]. LEIS and DRS are complementary techniques, performed simultaneously, with sensitivity to the masses of atoms in the outermost monolayers of a surface. LEIS and DRS are performed by directing a low energy ion beam ($He^+$ and $Ne^+$ for this work) at a grazing angle α onto the target surface. An electrostatic analyzer (ESA) is positioned in the forward scattering direction at a chosen scattering angle θ to detect the energies of scattered incident ions and recoiled target ions to perform LEIS and DRS, respectively. From the energies of the detected ions and the scattering angle θ, the mass of surface atoms can be calculated, yielding insight into the surface composition. One key consideration is that the ESA detects only scattered and recoiled ions; incident ions neutralized by the surface, as well as recoiled neutral atoms, are not detected. As such, neutralization effects can play a substantial role in the obtained ion energy spectra.

**X-ray absorption spectra (XAS)**. XAS measurements of boron K-edge were acquired at beamlines 7.3.1 and 8.0.1.4 at the Advanced Light Source (ALS), Lawrence Berkley National Laboratory (LBNL) and the REIXS beamline of the Canadian Light Source (CLS). Data were simultaneously collected in surface-sensitive total electron yield (TEY) measured as the drainage current from the sample and bulk-sensitive fluorescence yield (FY) measured using a channeltron electron multiplier. Spectra were normalized by dividing by incident beam intensity, which is monitored by drainage current from a gold grid located in the path of the X-ray beam immediately before the measurement chamber. A linear background was subtracted, and the absorption edge step was scaled to 1, taking the difference between pre-edge (<180 eV) and post-edge (>215 eV). The energy shift between different beamlines was calibrated using the primary peak of a $B_2O_3$ standard. Samples were transferred into the beamlines using vacuum suitcases (ALS) or a glove bag (CLS) to minimize exposure to atmospheric oxygen and moisture. The operating pressure of the endstations is better than $1 \times 10^{-9}$ Torr.

## Data availability

The DFT-optimized structures generated in this study are provided in the Supplementary Information/Source Data file. Details of computational models and additional computational and experimental characterization results are included in the Supplementary Information/Supplementary Information pdf file. Source data are provided with this paper.

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

## Acknowledgements

The authors acknowledge financial support through the Hydrogen Storage Materials Advanced Research Consortium (HyMARC) of the U.S. Department of Energy (DOE), Office of Energy Efficiency and Renewable Energy, Fuel Cell Technologies Office under contract DE-AC52-07NA27344. This work was performed under the auspices of the DOE by Lawrence Livermore National Laboratory (LLNL) under Contract DE-AC52-07NA27344. Sandia National Laboratories is a multimission laboratory managed and operated by National Technology and Engineering Solutions of Sandia, LLC., a wholly owned subsidiary of Honeywell International, Inc., for the U.S. Department of Energy's National Nuclear Security Administration under contract DE-NA-0003525. We thank Josh Whaley for his assistance with ion-scattering experiments and Dr Vincenzo Lordi for illuminating discussions on XAS simulations. Computing resources were provided under the LLNL Institutional Computing Grand Challenge program.

The views and opinions of the authors expressed herein do not necessarily state or reflect those of the United States Government or any agency thereof. Neither the United States Government nor any agency thereof, nor any of their employees, makes any warranty, expressed or implied, or assumes any legal liability or responsibility for the accuracy, completeness, or usefulness of any information, apparatus, product, or process disclosed, or represents that its use would not infringe privately owned rights.

## Author contributions

S.L. conceived the project. S.L., V.S. and B.C.W. coordinated the project. K.G.R. performed the XAS simulations. S.L., P.X., S.K. performed the atomistic simulations. C.-S.W. performed the LEIS experiments, with assistance from R.D.K. H.G. performed the synthesis of MgB2 materials, XRD, and EDS experiments. J.D.S. and C.D. performed the TEM experiments, and R.W.F. collected the AFM images. H.G., Y.-S.L., A.B., and J.L. performed the XAS experiments. S.L., H.G., C.-S.W., K.G.R., and B.C.W. wrote the manuscript and prepared the figures with help from the other co-authors. All the authors including L.F.W., J.J.U., K.J., and M.D.A. contributed to the scientific discussions, data analysis, and preparation of the manuscript.

## Competing interests

The authors declare no competing interests.
