## [Peer Review File · Nature Communications]

Spontaneous dynamical disordering of borophenes in MgB_2 and related metal boridesREVIEWER COMMENTS

Reviewer #1 (Remarks to the Author):

Review of the manuscript, "Spontaneous Dynamical Disordering of Borophenes in MgB₂ and Related Metal Borides", by Sichi Li et al.

This paper provides a practical surface borophene structure model of MgB₂ (0001) by VASP, a state-of-the-art electronic structure code, with comparing the experimental results. The type of research effort described in this manuscript is potentially of great importance for reconsidering what is a real surface structure apart from an idealized surface model. This work might be a groundbreaking work, and deserve to be published in this journal.

The manuscript (and supporting information provided) is very well written and organized. The numerical simulations including AIMD, CI-NEB, and metadynamics seems nice without any calculation condition faults and no inconsistency is found. The derived theoretical data are well-compared with the experiments, too.

I have only one request for improving the manuscript as below.

1. Please add a section or paragraph for justifying more the present model with a stoichiometric number of B atoms in the view of surface chemical composition.

In this work, the number of boron atoms (#B) per surface area seems identical, i.e., 32 B atoms on a 4x4 lattice (and 8 B atoms in a 2x2 surface lattices).

In my opinion, readers may be skeptical to the present results based on this "fixed number of atoms" calculation.

In general, a real surface is essentially open for the particle numbers, so that, #B can vary. Is there any possibility that a more stable borophene structure with a different #B?

For example, as a very simple one, let's consider a real hydrogen gas (i.e., Avogadro numbers of hydrogen exist) structure optimization problems with use of models with 3 and 4 hydrogens in a periodic boundary cell (PBC). The former calculation of 3 hydrogens must show that an atomic H and a H₂ molecule states must be stable where each H can be exchanged with some probability, while the latter should exhibit very stable two hydrogen molecules. Which does show the real situation? In this example, the former, 3 hydrogens model, should be apparently unmatched to reproduce the normal situation.

Thus, the real system containing many atoms can allow atoms go in and out in an interested local area, while a conventional simulation PBC cell should not permit it

I am afraid that the same situation may occur in the present surface simulation.

In other words, the number of surface boron atoms in the simulation cell should not be fixed for exploring the global minimum surface structure because boron (and Mg) may come from/go out to substrate or the other surface area easily by diffusion in the real in order to keep the chemical potential constant in a thermodynamic view.

I recommend the authors to check the effect of surface chemical composition fluctuation to the "dynamical disordered borophene structure" provided here since the authors focus on the practical/real (not-ideal) surface of MgB₂ in this article.

I think that it is essential to explain/justify why the discussion based on only the "32B/4x4 models, i.e., the stoichiometric numbers of B atoms, are OK to discuss in this article.

Reviewer #2 (Remarks to the Author):

In this manuscript by Li et al., the authors present a thorough study combining primarily computational simulations and characterization to show that the outer boron surface in MgB₂ nanosheets undergo disordering and clustering. This work is timely given the recent interest in

boron-based materials, and features well-conducted theory and experiment that have very interesting findings in terms of the resulting disordered structures that deviate from the hexagonal structure. The calculations regarding interactions with hydrogen are also interesting, and suggest possible relevance to applications. Overall, the paper is also well written and well organized.

However, the following issues need to be addressed before the work can be published:

1. In the experimentally prepared MgB₂ nanosheets, how were the conditions chosen? Why was acetonitrile chosen for the initial dispersion, but toluene for subsequent redispersion for TEM and AFM imaging?
2. Can the authors do HRTEM or some other direct imaging to show the reconstruction or amorphization of the surfaces? The measurements that are shown are primarily ensemble measurements that show the presence of certain elements and the lack of certain features (e.g. the B-B hexagonal bond), but do not directly show what the actual reconstructed surface looks like.
3. What is the effect of ball milling on the surface structures and how does it relate to a naturally occurring dynamical reconfiguration? The computational component suggests that at room temperature or slightly elevated temperatures that there can be ready movement between different disordered configurations, but the experiments with high energy ball milling are quite different. It would be good to show additional experimental evidence of what is happening during the exfoliation and dispersion processes.
4. What is the large sharp peak, and the smaller oscillations in Fig. 5a?
5. In the section related to hydrogen storage, the authors should first describe how hydrogen storage is expected to work, and what role the MgB₂ or other materials would play, in what way, and what kind of performance metrics are important. This would help to put their findings into context better. Are the heterogeneous sites on the disordered MgB₂ beneficial for hydrogen storage or not?
6. Can the authors discuss how their findings for surface disorder across the range of metal diborides (Fig. 7) might be related to recent experimental findings of liquid phase dispersion of nanosheets of several of the same compounds, and also of boron carbide? These were more recent 2021 works that were perhaps missed by the authors.
7. Do the XRD and XAS and other measurements primarily indicate that the outer B surfaces have changed away from the hexagonal structure, or do they also indicate that the resulting structures match the calculated LM₂, GM, etc. structures? Can the authors comment more conclusively on what the structures are likely to be in reality? Perhaps some mixture and dynamically changing configurations?

Minor comments:

- Do the labels LM₂, LM₃, and GM₁, etc., stand for local minimum and global minimum? This should be clearly defined.
- Line 351: Ref. [29] is plain text, not formatted citation.
- The title uses the term "borophenes", but I don't think this comes up very much in the main text in terms of the actual results, as opposed to when mentioning previous literature. I would probably suggest rewording the title, or alternately defining the term more clearly and using it more consistently throughout the rest of the paper.

Reviewer #3 (Remarks to the Author):

The authors report their extensive DFT calculations and ab initial MD simulations about a boron-terminated MgB₂ surface. They also report the related experimental synthesis and characterization of MgB₂ nanosheets with TEM, AFM, X-ray, LEIS, ARIES, XAS, and so on. The results, particularly the novel prediction and observation of disordering of boron surfaces, are very interesting. The work has great potential for various applications. The manuscript is also well-written. Thus, I would recommend an acceptance.

Response to referees letter

Reviewer 1

This paper provides a practical surface borophene structure model of MgB₂ (0001) by VASP, a state-of-the-art electronic structure code, with comparing the experimental results.

The type of research effort described in this manuscript is potentially of great importance for reconsidering what is a real surface structure apart from an idealized surface model.

This work might be a groundbreaking work, and deserve to be published in this journal.

The manuscript (and supporting information provided) is very well written and organized.

The numerical simulations including AIMD, CI-NEB, and metadynamics seems nice without any calculation condition faults and no inconsistency is found. The derived theoretical data are well-compared with the experiments, too.

Response: We appreciate the reviewer's positive comments on our manuscript.

I have only one request for improving the manuscript as below.

1. Please add a section or paragraph for justifying more the present model with a stoichiometric number of B atoms in the view of surface chemical composition.

In this work, the number of boron atoms (#B) per surface area seems identical, i.e., 32 B atoms on a 4x4 lattice (and 8 B atoms in a 2x2 surface lattices).

In my opinion, readers may be skeptical to the present results based on this "fixed number of atoms" calculation.

In general, a real surface is essentially open for the particle numbers, so that, #B can vary. Is there any possibility that a more stable borophene structure with a different #B?

For example, as a very simple one, let's consider a real hydrogen gas (i.e., Avogadro numbers of hydrogen exist) structure optimization problems with use of models with 3 and 4 hydrogens in a periodic boundary cell (PBC). The former calculation of 3 hydrogens must show that an atomic H and a H₂ molecule states must be stable where each H can be exchanged with some probability, while the latter should exhibit very stable two hydrogen molecules. Which does show the real situation? In this example, the former, 3 hydrogens model, should be apparently unmatched to reproduce the normal situation.

Thus, the real system containing many atoms can allow atoms go in and out in an interested local area,

while a conventional simulation PBC cell should not permit it
I am afraid that the same situation may occur in the present surface simulation.
In other words, the number of surface boron atoms in the simulation cell should not be fixed for exploring the global minimum surface structure because boron (and Mg) may come from/go out to substrate or the other surface area easily by diffusion in the real in order to keep the chemical potential constant in a thermodynamic view.

I recommend the authors to check the effect of surface chemical composition fluctuation to the “dynamical disordered borophene structure” provided here since the authors focus on the practical/real (not-ideal) surface of MgB₂ in this article.

I think that it is essential to explain/justify why the discussion based on only the “32”B/4x4 models, i.e., the stoichiometric numbers of B atoms, are OK to discuss in this article.

Response: We thank the reviewer for the insightful comment on the potential effect of surface composition fluctuation on dynamical disordered borophene structures. In fact, the effect of surface composition was studied by Liu *et al.* in the context of chemical-vapor deposition, [ref. 13 in the MS, DOI:10.1002/anie.201207972] where they computed borophene chemical potentials as a function of boron vacancy concentration x in B_{1-x}V_x ranging from $x=0$ for a fully packed boron sheet without vacancies to $x=1/3$ corresponding to a boron sheet with a stoichiometric number of B atoms with respect to the subsurface Mg. All borophenes were computed based on crystalline (pristine) structural models. They reported the borophene with a vacancy concentration x of 2/9 (B_{7/9}V_{2/9}) is the lowest in chemical potential, which is ~ 0.3 eV/B_{borophene} lower compared to the stoichiometric borophene (B_{2/3}V_{1/3}).

In the same spirit, we performed a series of new calculations to similarly determine the energies of borophenes at different vacancy concentrations (B_{1-x}V_x), but without constraining the structures to be in a crystalline (pristine) state. In addition to $x=1/3$, we included six additional vacancy concentrations x ranging from 1/12 to 5/12 (represented by 2x2, 3x3 or 4x4 surface models). For the 2x2 surface models, we optimized all enumerated symmetry-distinct initial structures. For 3x3 and 4x4 models, because full enumeration is computationally intractable, we explored local minima by basin-hopping optimizations starting from their pristine crystalline states. Lowest-energy structures for each x were used for subsequent calculations of their formation energies referenced to bulk α -B and clean Mg surface of MgB₂ according to the following equation:

$$E_B^f = (E_{B/Mg \text{ surface}} - E_{Mg \text{ surface}} - N_B E_{\alpha-B}) / N_B$$

where $E_{borophene/Mg \text{ surface}}$, $E_{Mg \text{ surface}}$, $E_{\alpha-B}$ are DFT-computed energies of B-terminated MgB₂ slab, Mg-terminated MgB₂ slab, and bulk α -B normalized to per B atom. N_B is the number of B atoms in the B-terminated MgB₂ surface.

Results including optimized borophene structures and $E_{\text{borophene}}^f$ are shown in the figure above. Key findings include: (1) depositing boron to form borophene on Mg surface of MgB_2 using α -B as the boron source is endothermic regardless of boron vacancy concentration x of the resulting borophenes; (2) borophenes with $x > \sim 0.275$ all exhibit tendency towards disordering; (3) $\text{B}_{7/9}\text{V}_{2/9}$ is still the lowest-energy borophene polymorph, but its energetic difference with respect to stoichiometric $\text{B}_{2/3}\text{V}_{1/3}$ is significantly reduced to ~ 0.1 eV/ $\text{B}_{\text{borophene}}$ from ~ 0.3 eV/ $\text{B}_{\text{borophene}}$ reported by Liu *et al.* (in which $\text{B}_{2/3}\text{V}_{1/3}$ was approximated using the pristine hexagonal configuration). It is expected that the energetic difference between $\text{B}_{7/9}\text{V}_{2/9}$ and $\text{B}_{2/3}\text{V}_{1/3}$ will be further reduced with larger surface models which allow higher configurational degree of freedom for geometric relaxation of disordered $\text{B}_{2/3}\text{V}_{1/3}$.

These results suggest that with high boron chemical potential, it is likely that boron atoms will form ordered patterns similar to those proposed by Liu *et al.* On the other hand, our results are more relevant at low boron chemical potentials, for which global enrichment of surface boron is unlikely, and disordering will be energetically preferred. It is also possible that under these conditions, surfaces will segregate into locally ordered B-enriched and disordered B-deficient regions. Note that these different ranges of boron chemical potential may correspond to different reaction conditions: for example, high chemical potentials will be observed during chemical-vapor deposition, whereas low chemical potentials will be dominated during exfoliation. Similarly, under reactive hydrogenation conditions, there is no additional native boron source, and one can expect a low relative boron chemical potential. Furthermore, atomic disordering is expected to introduce excess configurational entropy to the boron surface, which may further stabilize the disordered high-vacancy states. It is challenging to determine the magnitude of this configurational entropy; however, comparisons between calorimetry of glasses and crystals suggest that this could be in the range of 26 J/K mol at 298 K (based on data for crystalline and glass GeO_2 phases [doi:

10.1007/BF00209228]). Notably, this is large enough to alter thermodynamic stability under a broader range of chemical potentials.

In fact, we recently discovered that the disordering tendency of borophene sheets on MgB_2 under conditions with low boron chemical potential is supported by a recent study on MgB_2 atomic surface arrangement using scanning tunneling microscopy (STM), in which Sugimoto *et al.* reported their observation that boron atoms of a freshly cleaved MgB_2 are randomly distributed and irregularly arranged. [doi: 10.1063/10.0000132]. In short, we would like to thank the reviewer again for encouraging us to explore the effect of surface composition fluctuation, which allowed us to gain more insights into the underlying thermodynamics and more rigorously define appropriate conditions under which surface disordering is expected. We added two paragraphs (on page 5 and page 8) in the revised MS to discuss these new results and included all details as a new section in the revised SI (section 3). We also updated the language throughout the paper (abstract, results, and discussion sections) to qualify the conditions under which disordering is expected.

Page 5: “Depending on the boron chemical potential, addition or depletion of surface boron atoms can result in a variety of different patterns. Several of these were previously explored in first-principles calculations reported by Liu *et al.* in the context of chemical-vapor deposition synthesis, which corresponds to high relative chemical potential.¹⁸ Here, we are particularly interested in lower boron chemical potentials, in which the surface boron density is constrained to the stoichiometric composition. These conditions are broadly representative of exfoliation processes, as well as reaction conditions such as hydrogenation, for which there is no additional native boron source.”

Page 8: “We point out that the effect of surface composition was previously studied by Liu *et al.* for a range of boron chemical potentials beyond the stoichiometric composition, as represented by varying the surface boron density.¹⁸ They reported that for a boron-enriched MgB_2 surface can adopt an ordered boron configuration, which represents the global minimum in the zero-temperature formation energy. Because our calculations in Figures 1 did not explicitly explore different boron densities, we performed another series of calculations to explore surface patterns at other boron chemical potentials. Details are reported in SI section 3. Our results confirm that with high boron chemical potential, boron atoms will likely form ordered patterns similar to those proposed by Liu *et al.* On the other hand, surface disordering remains preferred at lower boron chemical potentials, including for both stoichiometric and substoichiometric (boron-depleted) surfaces. This in turn introduces the possibility that at intermediate boron chemical potentials, surfaces will segregate into locally ordered B-enriched and disordered B-deficient regions. Note that these different ranges of boron chemical potential may correspond to different reaction conditions: for example, high chemical potentials will be observed during chemical-vapor deposition (the application of interest within the study of Liu *et al.*), whereas low chemical potentials will dominate during processes such as exfoliation. Similarly, under reactive hydrogenation conditions (explored further below), there is no additional native boron source, and one can expect a low relative boron chemical potential. Furthermore, atomic disordering is expected to introduce excess configurational entropy to the boron surface, which may further stabilize the disordered high-vacancy states.”

Other minor changes are highlighted in the diff.pdf file.

Reviewer2

In this manuscript by Li et al., the authors present a thorough study combining primarily computational simulations and characterization to show that the outer boron surface in MgB₂ nanosheets undergo disordering and clustering. This work is timely given the recent interest in boron-based materials, and features well-conducted theory and experiment that have very interesting findings in terms of the resulting disordered structures that deviate from the hexagonal structure. The calculations regarding interactions with hydrogen are also interesting, and suggest possible relevance to applications. Overall, the paper is also well written and well organized.

Response: We thank the reviewer for taking the time to evaluate our manuscript and for the positive comments.

However, the following issues need to be addressed before the work can be published:

1. In the experimentally prepared MgB₂ nanosheets, how were the conditions chosen? Why was acetonitrile chosen for the initial dispersion, but toluene for subsequent redispersion for TEM and AFM imaging?

Response: In this work, MgB₂ nanosheets were synthesized via solid-state exfoliation route by using high energy ball mill. To separate the exfoliated nanosheets from the unexfoliated fraction, we dispersed the milled powder in various commonly available organic solvents. We observed that the exfoliated MgB₂ nanosheets remain as stable dispersion in solvents like Acetonitrile, Isopropanol, Dimethylformamide, and Ethanol. To obtain the nanosheets in powder form, we use the freeze-drying process and acetonitrile has a freezing point of -45 °C and a boiling point of 82 °C, which makes it feasible to operate in the lyophilizer. Thus, to recover the nanosheets in powder form, we used acetonitrile during the separation process. While performing the microscopic imaging like TEM and AFM, the MgB₂ nanosheets were redispersed in toluene because this solvent prevents excessive agglomeration of the nanosheets. In addition, the toluene dispersion of MgB₂ remains stable even with a low concentration of nanosheets on the substrate, which helps in getting better quality images. We added sentences in the methodology section to describe the rationale of conditions chosen:

Page 28, “We chose ACN as it has a freezing point of -45 °C and a boiling point of 82 °C, which makes it feasible to operate in the lyophilizer in the subsequent synthesis procedure.”

Page 29, “Samples for TEM analysis were prepared by taking a small amount of MgB₂ nanosheets powder and dispersed in 10 ml of dry toluene, to prevent excessive agglomeration of the nanosheets, using a bath ultrasonicator for 5 minutes.”

2. Can the authors do HRTEM or some other direct imaging to show the reconstruction or amorphization of the surfaces? The measurements that are shown are primarily ensemble measurements that show the presence of certain elements and the lack of certain features (e.g. the B-B hexagonal bond), but do not directly show what the actual reconstructed surface looks like.

Response: We thank the reviewer for kindly suggesting performing HRTEM imaging of the nanosheets. Unfortunately, the HRTEM measurements on solvent-exfoliated MgB₂ nanosheets are inconclusive. Since TEM measurements are done in transmission mode which is not surface-sensitive, it is difficult to discern the surface features. The HRTEM images we obtained show seemingly amorphous regions which we believe are due to the presence of multiple MgB₂ grains, which is also obvious from the selected area electron diffraction patterns below.

We turned to the literature instead, and found a recent study on MgB₂ atomic surface arrangement using scanning tunneling microscopy (STM), which is a surface-sensitive technique, published by Sugimoto *et al.* [*Fiz. Nizk. Temp.* 45, 1423–1433 (2019); doi: 10.1063/10.0000132]. The paper provides detailed information on surface atomic arrangement in magnesium diboride and site occupancy. The relevant figure is shown below.

Editorial Note: Reprinted from [Akira Sugimoto, Yuta Yanase, Toshikazu Ekino, Takahiro Muranaka, and Alexander M. Gabovich, "Atomic structures and nanoscale electronic states on the surface of MgB₂ superconductor observed by scanning tunneling microscopy and spectroscopy", *Low Temperature Physics* 45, 1209-1217 (2019)], with the permission of AIP Publishing.

FIG. 1. (a) The STM image (topography) of MgB₂ with the atomic arrangement obtained at the $T = 4.9$ K ($V = +15$ mV, $I = 0.3$ nA). (b) The fast Fourier transformed (FFT) power spectrum image of the STM image of Fig. 1(a). (c) The crystal structure of MgB₂ with the atomic lattice lengths. (d) The schematic draw of the atomic arrangement on the observed area together with its hexagonal structures and the STM contour plot. (e) The schematic draw of the Mg atom location (red dots) derived from Fig. 1(d), together with the FFT power spectrum image. (f) Two types of the representative conductance (dI/dV) STS spectra for MgB₂ obtained at $T = 4.9$ K.

Shown in FIG 1(d) of [*Fiz. Nizk. Temp.* 45, 1423–1433 (2019); doi: 10.1063/10.0000132], the Mg atoms (in red) probed by STM in MgB₂ at 4.9 K show the periodic Mg–Mg bond lengths (0.260–0.266 nm), consistent with the Mg atomic structure. In contrast, the boron atoms (in blue) seem to be “...*randomly distributed and irregularly arranged in the STM image manifesting themselves as weak peaks*”, corroborating our predictions. We included a citation to the paper by Sugimoto *et al.* in the revised version of the MS (ref # 34). We also revised the text of the MS as follows:

Page 18, “However, Sugimoto *et al.* similarly observed through scanning tunneling microscopy that boron atoms on a freshly-cleaved MgB₂ sheets are randomly distributed and irregularly arranged even in the

absence of ball milling.³⁴ These results confirm that surface disordering also occurs under nonaggressive synthetic conditions.”

3. *What is the effect of ball milling on the surface structures and how does it relate to a naturally occurring dynamical reconfiguration? The computational component suggests that at room temperature or slightly elevated temperatures that there can be ready movement between different disordered configurations, but the experiments with high energy ball milling are quite different. It would be good to show additional experimental evidence of what is happening during the exfoliation and dispersion processes.*

Response: The reviewer raises a good point. Here, we used high energy ball milling technique for nanoscaling MgB₂ to obtain the nanosheets. Ball milling is a shear-force dominant process where the particle size goes on reducing by impact and attrition of grinding media. In this process, MgB₂ particles are believed to break down by randomly striking with grinding media in the rotating shell to create shear and compression force which helps overcome the weak Van der Waals interactions between the boride layers. The high-energy milling also breaks down the material in the lateral dimension, creating smaller particles. Ball milling also enhances the surface area, and our approach was to increase the concentration of freshly formed surfaces to have a better chance of capturing this purely surface dynamical reconfiguration phenomenon. Our low energy ion scattering spectra provide two indications that these surfaces are primarily boron-terminated: the spectrum obtained with a Ne beam contains a large peak corresponding to recoiled boron atoms, while the peaks in the spectrum obtained with a He beam are greatly suppressed due to a matrixing (neutralization) effect. We cannot rule out the possibility that ball-milling enhances the kinetics of surface disordering, but we note that some surface disordering is observed even without ball-milling, as shown by Sugimoto *et al.* for freshly-cleaved MgB₂ surfaces [*Fiz. Nizk. Temp.* 45, 1423–1433 (2019); doi: 10.1063/10.0000132]. We added sentences in the revised MS as follows to inform readers of the potential effect of high energy ball milling, and the literature evidence that surface disordering is also observed under nonaggressive synthetic conditions:

Page 18, “We acknowledge the possibility that ball-milling enhances the kinetics of surface disordering within our samples. However, Sugimoto *et al.* similarly observed through scanning tunneling microscopy that boron atoms on a freshly-cleaved MgB₂ sheets are randomly distributed and irregularly arranged even in the absence of ball milling.³⁴ These results confirm that surface disordering also occurs under nonaggressive synthetic conditions.”

4. *What is the large sharp peak, and the smaller oscillations in Fig. 5a?*

Response: As described by Ray *et al.* [doi: 10.1039/c7cp03709k], the large sharp peak (193.7 eV) is originated from planar three-fold coordinated boron atoms, and both MgB₂ (boron surrounded by three other boron atoms) and B₂O₃ (boron surrounded by three oxygen atoms) can potentially exhibit such feature. Small oscillations, for instance in B₂O₃ spectrum, are mainly noise.

5. *In the section related to hydrogen storage, the authors should first describe how hydrogen storage is expected to work, and what role the MgB₂ or other materials would play, in what way, and what kind of*

performance metrics are important. This would help to put their findings into context better. Are the heterogeneous sites on the disordered MgB₂ beneficial for hydrogen storage or not?

Response: MgB₂ can absorb hydrogen and form Mg(BH₄)₂, a material which has one of the highest gravimetric hydrogen capacities of any metal hydrides, about 14.9 wt.%. However, the MgB₂-Mg(BH₄)₂ H₂ storage system suffers from sluggish kinetics of hydrogenating MgB₂ to Mg(BH₄)₂, necessitating high temperatures (350-400 °C) in hydrogenation half cycles. As being pointed out at the beginning of the surface reactivity section (page 18), our findings of disordered boron surface shed light on the kinetic aspect of MgB₂-to-Mg(BH₄)₂ phase transformation:

“Liu et al. proposed that the intrinsic stability of the B–B extended hexagonal ring structure hinders the overall hydrogenation kinetics of MgB₂ (with an underlying assumption that B maintains hexagonal patterns even when surface-exposed).³³ As such, it is critical to understand which factors can facilitate B–B bond disruption. The current study suggests that such disruption is achievable natively at surfaces due to surface disordering, prompting a more direct investigation into the implications for hydrogenation.”

Further, we discovered that exposed disordered boron surface of MgB₂ contains heterogeneous surface sites different to the pristine surface, and importantly among them there are surface-B-subsurface-Mg sites that exhibit significantly lower kinetic barrier for H₂ activation. Therefore, nanostructuring and exfoliation of MgB₂ to maximize the fraction of boron-terminated surfaces should in theory result in faster kinetics of initial hydrogenation of MgB₂. We added descriptions in the revised MS to provide more background of MgB₂ as hydrogen storage materials, and to highlight the potential benefit of exposing disordered boron surfaces for hydrogen storage:

Page 18, “MgB₂ can absorb hydrogen and form Mg(BH₄)₂, a material which has one of the highest gravimetric hydrogen capacities of any metal hydrides, about 14.9 wt.%. However, the MgB₂-Mg(BH₄)₂ H₂ storage system suffers from sluggish kinetics of hydrogenating MgB₂ to Mg(BH₄)₂, necessitating high temperatures (350-400 °C) in hydrogenation half cycles.”

Page 20, “However, dissociation on the surface B and subsurface Mg sites is extremely facile (0.38 eV), suggesting that nanostructuring to maximize boron-surface-area-to-volume ratio is beneficial for hydrogenation kinetics of MgB₂.”

6. Can the authors discuss how their findings for surface disorder across the range of metal diborides (Fig. 7) might be related to recent experimental findings of liquid phase dispersion of nanosheets of several of the same compounds, and also of boron carbide? These were more recent 2021 works that were perhaps missed by the authors.

Response: We thank the reviewer for informing us of recent findings in the field. We were able to find two relevant publications in 2021 by Yousaf *et al.* [[doi: 10.1021/acs.jpcc.1c00394](https://doi.org/10.1021/acs.jpcc.1c00394)], and Guo *et al.* [[doi: 10.1039/D0NR07971E](https://doi.org/10.1039/D0NR07971E)], respectively: one on liquid phase dispersion of nanosheets of a wide range of metal diborides including MgB₂, and the other on ultrathin nanosheets of boron carbide. In the publication

by Yousaf *et al.*, they confirmed the feasibility of synthesizing nanosheets of AlB₂, CrB₂, HfB₂, MgB₂, NbB₂, TaB₂, TiB₂, and ZrB₂. Most of these compounds are included in our work (Figure 7 in the MS) and do not exhibit boron surface disordering except for MgB₂. Yousaf *et al* reported HRTEM images of HfB₂ and MgB₂, and their MgB₂ images (Figure S7b) look similar to the HRTEM images measured by us. However, as mentioned in the response above, due to the transmission nature, HRTEM do not allow conclusive validation of our predicted single-layer surface phenomena. Nonetheless, their report of successful synthesis of various metal diboride nanosheets promises future work of using surface-sensitive characterization, e.g. STM, to validate our predictions. In another paper by Guo *et al.*, they reported their synthesis of boron carbide (B₄C) nanosheets through liquid phase exfoliation. Interestingly, they observed, through Raman spectroscopy, atomic rearrangements and amorphization at the surfaces of B₄C nanosheets, similar to boron surface disordering of MgB₂. We included these references and discussed their relevance to our findings in the revised MS:

Page 22, “Notably, Yousaf *et al* confirmed the feasibility of synthesizing nanosheets for most of these metal diborides through liquid phase dispersion,³⁷ allowing future experimental validations of our predictions.”

Page 22, “In addition, surface atomic rearrangements and amorphization were also observed by Guo *et al.* for boron carbide nanosheets through Raman spectroscopy,⁴⁰ suggesting that similar physiochemical phenomena may be relevant for a wider range of nanoscale materials.”

7. Do the XRD and XAS and other measurements primarily indicate that the outer B surfaces have changed away from the hexagonal structure, or do they also indicate that the resulting structures match the calculated LM2, GM, etc. structures? Can the authors comment more conclusively on what the structures are likely to be in reality? Perhaps some mixture and dynamically changing configurations?

Response: Strictly speaking, the XRD and XAS measurements are a combination of the signal obtained surface and bulk states. In XRD the penetration depth is rather large (several microns), so the signal is dominated by bulk, therefore no surface information can be obtained. In XAS, the total fluorescence yield (TFY) mode has a sensitivity of hundreds of nm, whereas measurements in the total electron yield (TEY) XAS mode offer the ability to separately probe surface phenomena. However, while XAS in TEY mode clearly indicate that the outer B surfaces have changed away from the hexagonal structure, evidenced by our XAS simulations, further resolving the exact atomic surface structure based on XAS spectra remains challenging. Moreover, exactly as the reviewer suggests, we envision that in reality surface structures should be amorphous with dynamically changing configurations, so what XAS is probing should be a geometric and time ensemble of surface structures. We added a sentence to comment more on in reality what the structures are likely to be in the revised MS:

Page 17, “As borophenes on MgB₂ are predicted to be amorphous with dynamically changing configurations, we propose that experimental XAS is actually probing a geometric ensemble and time average of surface structures. Nonetheless, the pristine borophene and LM4 surface boron patterns should inform qualitatively the effect of atomic disordering on the spectral line shapes.”

Minor comments:

- Do the labels LM2, LM3, and GM1, etc., stand for local minimum and global minimum? This should be clearly defined.

Response: LM and GM in our original manuscript stand for local minimum and global minimum, respectively. On reflection, in the spirit of the previous comment and response, we think it is more rigorous to term all disordered surface configurations as local minima since the surface configurational space is naturally a function of modeled surface size and GM1 is at most the global minimum within a 4x4 surface model. For this reason, we decided to change GM1 to LM4 and define clearly what LM stands for in the revised manuscript.

Page 7, “Top-view structures of the two iso-energetic surfaces with the lowest energy from our initial search (LM2, LM3, **where LM stands for local minimum**) are shown alongside the pristine surface (LM1) in Figure 1.”

- Line 351: Ref. [29] is plain text, not formatted citation.

Response: We fixed the formatting.

- The title uses the term "borophenes", but I don't think this comes up very much in the main text in terms of the actual results, as opposed to when mentioning previous literature. I would probably suggest rewording the title, or alternately defining the term more clearly and using it more consistently throughout the rest of the paper.

Response: We thank the reviewer for suggesting using consistent terminology. Borophene by definition is a crystalline atomic monolayer of boron. Strictly speaking, only surface boron with hexagonal pattern belongs to the borophene family and disordered surface boron layers do not. To be rigorous, we included the definition of borophene and used it to term the pristine hexagonal boron surface in the result section of the revised MS.

Page 4, “**Borophene by conventional definition is a crystalline atomic monolayer of boron.** In this work, we revisit the question of surface boron structure by combining DFT calculations with an exhaustive global optimization approach and free energy sampling.”

Other minor changes are highlighted in the diff.pdf file.

Reviewer3

The authors report their extensive DFT calculations and ab initial MD simulations about a boron-terminated MgB₂ surface. They also report the related experimental synthesis and characterization of MgB₂ nanosheets with TEM, AFM, X-ray, LEIS, ARIES, XAS, and so on. The results, particularly the novel prediction and observation of disordering of boron surfaces, are very interesting. The work has great potential for various applications. The manuscript is also well-written. Thus, I would recommend an acceptance.

Response: We thank the reviewer for the positive recommendation.

REVIEWERS' COMMENTS

Reviewer #1 (Remarks to the Author):

The authors have addressed all my comments in that justification of the present model should be required in the revised manuscript from the viewpoint of surface boron chemical potential. The updated manuscript includes new data and discussion related to the various surface B composition, and I think it can be satisfactory for potential readers of this article.

I would like to recommend the revised manuscript to be published in this journal.

Reviewer #2 (Remarks to the Author):

The authors have done an excellent job of responding to all the reviewers' comments, including adding some new experimental and computational work, and finding more relevant literature to support their findings. Overall, this is a very well-written and well-organized paper that has many intriguing and timely findings that should be published. This paper answers many important questions about nanoscale boron that will be of interest to the community working on these emerging related materials.

Response to referees

Reviewer #1 (Remarks to the Author):

The authors have addressed all my comments in that justification of the present model should be required in the revised manuscript from the viewpoint of surface boron chemical potential. The updated manuscript includes new data and discussion related to the various surface B composition, and I think it can be satisfactory for potential readers of this article.

I would like to recommend the revised manuscript to be published in this journal.

Response: We thank the reviewer for recommending our revised manuscript for publication.

Reviewer #2 (Remarks to the Author):

The authors have done an excellent job of responding to all the reviewers' comments, including adding some new experimental and computational work, and finding more relevant literature to support their findings. Overall, this is a very well-written and well-organized paper that has many intriguing and timely findings that should be published. This paper answers many important questions about nanoscale boron that will be of interest to the community working on these emerging related materials.

Response: We thank the reviewer for kind words on our manuscript.